# Efficient Automated Circuit Discovery in Transformers using Contextual Decomposition

**Aliyah R. Hsu**
Department of EECS
UC Berkeley
aliyahhsu@berkeley.edu

**Georgia Zhou**
Department of EECS
UC Berkeley

**Yeshwanth Cherapanamjeri**
CSAIL, MIT

**Yaxuan Huang**
Department of Statistics
UC Berkeley

**Anobel Y. Odisho & Peter R. Carroll**
Department of Urology, Epidemiology and Biostatistics
UC San Francisco

**Bin Yu**
Department of Statistics, EECS
Center for Computational Biology
UC Berkeley

## Abstract

Automated mechanistic interpretation research has attracted great interest due to its potential to scale explanations of neural network internals to large models. Existing automated circuit discovery work relies on activation patching or its approximations to identify subgraphs in models for specific tasks (circuits). They often suffer from slow runtime, approximation errors, and specific requirements of metrics, such as non-zero gradients. In this work, we introduce contextual decomposition for transformers (CD-T) to build interpretable circuits in large language models. CD-T can produce circuits at any level of abstraction and is the *first* to efficiently produce circuits as fine-grained as attention heads at specific sequence positions. CD-T is compatible to all transformer types, and requires no training or manually-crafted examples. CD-T consists of a set of mathematical equations to isolate contribution of model features. Through recursively computing contribution of all nodes in a computational graph of a model using CD-T followed by pruning, we are able to reduce circuit discovery runtime from hours to seconds compared to state-of-the-art baselines. On three standard circuit evaluation datasets (indirect object identification, greater-than comparisons, and docstring completion), we demonstrate that CD-T outperforms ACDC and EAP by better recovering the manual circuits with an average of 97% ROC AUC under low runtimes. In addition, we provide evidence that faithfulness of CD-T circuits is not due to random chance by showing our circuits are 80% more faithful than random circuits of up to 60% of the original model size. Finally, we show CD-T circuits are able to perfectly replicate original models' behavior (faithfulness = 1) using fewer nodes than the baselines for all tasks. Our results underscore the great promise of CD-T for efficient automated mechanistic interpretability, paving the way for new insights into the workings of large language models. All code for using CD-T and reproducing results is made available on Github. [1]

## 1 Introduction

Transformers (Vaswani et al., 2017) have recently demonstrated impressive predictive capabilities (Brown et al., 2020b) by learning intricate nonlinear relationships between features. How-

---

[1] https://github.com/adelaidehsu/CD_Circuit

ever, the challenge of comprehending these relationships has resulted in transformers largely being thought of as black boxes. Despite this, transformers are increasingly utilized in high-stakes domains such as medicine (e.g. medical image analysis (He et al., 2023)) and science (e.g. protein structure prediction aiding drug discovery (Jumper et al., 2021)). This highlights the necessity of understanding model behaviors. Mechanistic interpretability, a study to explain behaviors of neural networks in terms of their internal components, is at the frontier of interpretability research (Geiger et al., 2021; Geva et al., 2020; Räuker et al., 2023), as it uniquely provides an avenue for guiding manual modifications (Elhage et al., 2021; Vig et al., 2020b) and reverse-engineering solutions (Elhage et al., 2022; Meng et al., 2023).

One main direction in mechanistic interpretability research is circuit discovery, where researchers aim to identify a computational subgraph (circuit) in neural networks responsible for solving a specific task (Olah et al., 2020). Prior circuit discovery work in language models has found subgraphs of attention heads and multi-layer perceptrons (MLPs) that partially or fully explain model behaviors on tasks such as indirect object identification, docstring completion, and greater-than comparisons (Wang et al., 2023; Hanna et al., 2023; Nanda et al., 2023; Heimersheim & Janiak, 2023). However, most of them require tedious manual inspection, and are thus limited to smaller models (Lindner et al., 2023). Automated Circuit DisCovery (ACDC) (Conmy et al., 2023) has been proposed as a scalable end-to-end technique to identify circuits in models of arbitrary sizes by automating the pruning of unimportant edges via activation patching. Although effective, ACDC suffers from slow runtimes. More recent work has focused on improving the efficiency, for example, through linear approximating activation patching with attribution patching (Syed et al., 2023; Hanna et al., 2024), or training sparse autoencoders (SAEs) to learn attention head output patterns (O'Neill & Bui, 2024). While faster variants have been leveraged with some success, they still suffer from several drawbacks, such as approximation errors (Syed et al., 2023; Hanna et al., 2024), and the need of model training and carefully designed examples by human (O'Neill & Bui, 2024).

Our main contribution in this work is the introduction of contextual decomposition for transformers (CD-T), a mathematical or analytical method to efficiently compute the contribution of model features to other model features within a transformer. This work builds on CD methods for CNNs and RNNs to examine local importance proposed in prior work (Murdoch et al., 2018; Singh et al., 2018; Jumelet et al., 2019). We significantly extend CD by defining novel mathematical decomposition principles which can be applied to transformers of all types (e.g., decoder-based, encoder-based), because these principles are indispensable for CD to scale to modern state-of-the-art (SOTA) deep learning models as they are mostly based on transformers.

We are also the first, to our knowledge, to show the utility of CD methods for mechanistic interpretability by proposing an automated circuit discovery algorithm utilizing CD-T. That is, CD-T is able to discover circuits with finer granularity by further splitting into sequence positions while at the same time reduce runtime from hours to *seconds*, when it is compared with prior work (Syed et al., 2023; Conmy et al., 2023). CD-T is automatic or does not require any training or manually-crafted examples by humans. CD-T is fundamentally compatible with all common transformer architectures: in particular, CD-T supports circuit discovery in both BERT-like (Devlin et al., 2019) and GPT-like (Brown et al., 2020a) architectures.

Specifically, for evaluation, we use three standard circuit evaluation datasets: indirect object identification (IOI), greater-than comparisons (Greater-than), and docstring completion (Docstring). We compare recovered circuits by CD-T on these datasets with two SOTA baselines, ACDC (Conmy et al., 2023) and EAP (Syed et al., 2023). We demonstrate that CD-T outperforms the baselines by better identifying attention heads in the manual circuits with an average of 97% ROC AUC due to its outstanding ability in recovering negative and supporting heads efficiently. EAP obtains similar runtime as CD-T but performs slightly worse in ROC AUC for Greater-than and IOI. ACDC is the least efficient as it often takes hours to identify a circuit, yet with no better ROC AUC on all tasks compared to other methods. In addition, we demonstrate faithfulness of CD-T circuits is not due to random chance by showing our circuits are 80% more faithful than random circuits of up to 60% of the original model size. Finally, we show CD-T circuits are able to perfectly replicate original models' behavior (faithfulness = 1) using fewer nodes [2] than the baselines for all tasks.

---

[2] In our experiments, "nodes" refers to attention heads or attention heads' output at a specific sequence. See the general definition in Section 3.3.1.

## 2 RELATED WORK

### 2.1 CAUSAL INTERPRETATION

Previous efforts to interpret neural networks have frequently drawn on ideas from causal inference (Pearl, 2009). Much of this work emphasizes counterfactual reasoning or enforcing causal constraints on computations to explain model outputs (Pearl, 2009; Feder et al., 2020; Geiger et al., 2021; Wu et al., 2021; Kaddour et al., 2022). A related line of research (Vig et al., 2020a; Stolfo et al., 2023; Feng & Steinhardt, 2023) takes a different approach by treating internal components (or nodes, if the model is viewed as a computational graph) as mediators and conducting causal mediation analysis, also called *activation patching*. In these studies, the contribution of nodes is assessed by ablating subsets of them and measuring direct and indirect effects, with the direct effects serving as proxies for their overall contribution. In contrast to viewing nodes as mediators, Goldowsky-Dill et al. (2023) and Wang et al. (2023) propose treating input-to-output paths as more expressive mediators. They introduce the technique of *path patching* to explore how different edge subsets in a model's computational graph affect its behavior.

### 2.2 CIRCUIT DISCOVERY METHODS

Ablation-based methods are essential for identifying key components within models. Through activation patching, prior research on circuit discovery in language models has uncovered subgraphs of attention heads and MLPs that explain model behavior (Wang et al., 2023; Hanna et al., 2023; Heimersheim & Janiak, 2023). However, most of these methods require multiple iterations of manual inspection and ad-hoc analysis which is specific to the model and task, limiting their application to small numbers of circuits in smaller models (Lindner et al., 2023). To address this, Conmy et al. (2023) introduced Automated Circuit Discovery (ACDC), which scales to models of any size by automatically calculating edge importance between attention heads and MLPs based on model performance on a chosen metric via activation patching. While ACDC is effective, it suffers from slow runtimes and fails to recover certain attention head subsets within the manual circuits. Syed et al. (2023) introduced Edge Attribution Patching (EAP, or ACDC++ as later renamed), which improves ACDC's efficiency by using a linear approximation of activation patching. However, this reliance on linear approximations can lead to overestimation of edge importance and a weaker correlation with true causal effects. EAP also struggles when the gradient of the performance metric is zero. To resolve these issues, Hanna et al. (2024) proposed Edge Attribution Patching with Integrated Gradients (EAP-IG) for more faithful circuit recovery, although it increases runtime by a constant factor. Another approach trains sparse autoencoders (SAEs) on attention head outputs and uses the SAE-learned features to map attention head contributions to identified circuits. However, SAE-based methods either require carefully crafted human-generated examples (O'Neill & Bui, 2024) or require an impractical amount of compute to train an SAE (He et al., 2024; Marks et al., 2024). Another common limitation of prior methods is the limited level of granularity of the circuits they discover, because most work either doesn't support further attention head splitting at specific sequence positions (Syed et al., 2023; Hanna et al., 2024; O'Neill & Bui, 2024), or it is fundamentally feasible but requires a prohibitive amount of time to do so (Conmy et al., 2023), which drastically hinders more fine-grained mechanistic interpretability analysis. Our method CD-T addresses the limitations of previous circuit discovery approaches by significantly reducing runtime, using mathematical decomposition equations with no approximations, allowing for arbitrary level of abstraction of circuits, and avoiding both model training and manually crafted examples (i.e. CD-T achieves all the above in an automated manner).

### 2.3 CONTEXTUAL DECOMPOSITION IN NEURAL NETWORKS

Contextual Decomposition (CD), introduced by Murdoch et al. (2018), attributes local importance to input features in LSTMs by analytically decomposing the output without any changes to the underlying model. For each word in a sentence, a forward pass computes activations for all cells and gates while partitioning each neuron's activation into parts influenced by a selected token or phrase (denoted as the *relevant*) and those that are not (denoted as the *irrelevant*), using a factorization of the update equations for hidden states and cell states. Jumelet et al. (2019) propose Generalized Contextual Decomposition (GCD) for unidirectional LSTMs, applying it to phenomena like number agreement and pronoun resolution. Singh et al. (2018) extend CD from LSTMs to RNNs and

CNNs and introduce hierarchical interpretation with feature clustering for improved input feature attributions. A distantly related work, contextual explainability (Bertolini et al., 2022), focuses on feature attributions for each convolutional layer of CNNs by utilizing gradients as the feature importance measure. We further extend CD to transformers (CD-T) and we argue that our development of CD-T is a significant advance with two novel contributions. First, we carefully design mathematical decomposition principles tailored for transformers of all types (e.g., decoder-based, encoder-based), which makes it able to scale to larger models. Second, given prior work on CD only involves input feature attribution, to our knowledge, we are the first to demonstrate the utility of CD for mechanistic interpretability by proposing an automated circuit discovery algorithm that leverages CD-T.

## 3 OUR METHOD: CD-T AND MECHANISTIC INTERPRETABILITY

In this section, we first recall the basic operations of transformers (Section 3.1). We then provide technical details on contextual decomposition and its application to transformers (section 3.2). Finally, we conclude with a circuit discovery algorithm based on CD-T (section 3.3).

### 3.1 TRANSFORMERS

The main technical element unifying transformer-based architectures is the attention mechanism. This mechanism allows contextually useful information to be transmitted from one position in the sequence to any other position. We start by describing the operation of a general attention mechanism before discussing the modifications needed to specialize it for generation, exemplified by models such as GPT (Brown et al., 2020a), and for sequence-to-sequence transformation models such as BERT (Devlin et al., 2019). Formally, the attention mechanism computes a vector of contributions from a series of vectors $\{x_i\}_{i=1}^l$ to a target vector $x$ at position $p$ with $x_i, x \in \mathbb{R}^d$. An attention layer typically consists of a small number of heads, with each head parameterized by three functions, the key function $f_k$, value function $f_v$, and query function $f_q$. The output of the attention head to $x$ is computed as follows where $d_k$ denotes the output dimension of $f_k$:

$$\forall i \in [l] : k_i = f_k(x_i), q_x = f_q(x), v_i = f_v(x_i) \tag{1}$$

$$\forall i \in [l] : \alpha_i = \frac{\exp(q_x^\top k_i / \sqrt{d_k})}{\sum_{j \in [l]} \exp(q_x^\top k_j / \sqrt{d_k})} \tag{2}$$

$$y_x = \sum_{j=1}^l \alpha_i v_i. \tag{3}$$

The functions $f_k, f_v, f_q$ are commonly simple learnable transformations such as linear transformations or one hidden-layer MLPs. An attention layer is composed of a series of attention heads applied to every position in the sequence such that the concatenation of their outputs equals the input dimension $d$. As the output of an attention layer is a series of vectors of the same length and dimensionality as its input, this allows for stacking of these layers to build increasingly complex representations of the input so far.

The two classes of transformer models we consider in this work, sequence-to-sequence models such as BERT and generative models such as GPT mainly differ in the set of positions that the attention mechanism operates on. Concretely, BERT maps a *fully specified* input sequence to an output sequence of the sample length while generative models produce increasingly larger sequences autoregressively with the $(t + 1)^{th}$ token generated based on the previous $t$ tokens. Since *all* the input tokens are specified for BERT-based models, the computation of the attention vector at position $t$ utilizes representations at *every* position of the sequence. On the other hand, when generating the $(t + 1)^{th}$, the positions at $t + 1$ and beyond are not determined. As a consequence, only the representations for tokens before $t$ are used to compute the attention vectors.

### 3.2 CONTEXTUAL DECOMPOSITION FOR TRANSFORMERS (CD-T)

We will now describe the general method of contextual decomposition (CD), as well as our extension to the transformer architecture and applications to mechanistic interpretability.

CD (Murdoch et al., 2018) was first proposed to compute contributions of input tokens to the output of LSTMs as a local interpretation method. CD divides each cell and hidden state into a sum of two parts: a $\beta$ part (said to be *relevant*), which contains the part of this particular state that stems from the input tokens of interest, and a $\gamma$ part (said to be *irrelevant*), which contains information coming from tokens outside of the list of interest. $\beta$ is often initialized with masked input embeddings (1's specify where token positions of interest are, and 0's the opposite), and $\gamma$ the complement of $\beta$.

Given a decomposition of a vector $x \in \mathbb{R}^d$ into *relevant* and *irrelevant constituents* $\beta + \gamma = x$ (here we use the word *constituents* instead of "components" to avoid name collision with the "components of a neural network"), and a module $f : \mathbb{R}^d \to \mathbb{R}^k$, CD consists of a set of mathematical equations to determine the decomposition of the output of a module $f$ when given $x$ as input: $f(x) \in \mathbb{R}^k = \beta_o + \gamma_o$. The general principle is to find a symbolic expression for $f(x)$ in terms of $\beta$ and $\gamma$, and group the terms in the expression according to whether they are solely a function of $\beta$ or not: the sum of terms solely relying on $\beta$ becomes $\beta_o$, and the sum of the remaining terms becomes $\gamma_o$. We refer interested readers to Murdoch et al. (2018) and Singh et al. (2019) for further examples of decomposition formulas of various modules. These decomposition rules can be composed through multiple modules (i.e, if $f(\beta_1, \gamma_1) = \beta_2, \gamma_2$, and $g(\beta_2, \gamma_2) = \beta_3, \gamma_3$, then $g(f(\beta_1, \gamma_1)) = \beta_3, \gamma_3$) in order to define decomposition rules for larger computational blocks, including entire neural networks.

Other authors have heuristically adjusted the decomposition rules for a module to reflect specific mechanisms by which (the relevance term in an earlier layer $i - 1$) can affect (the relevance term in a later layer $i$) that are unaccounted for by this rule. For example, Singh et al. (2019) adjust the decomposition rule for affine transformations $f(x) = Wx + b$, so that the bias term does not affect the relative magnitudes of $\beta$ and $\gamma$:

$$\beta_i = W\beta_{i-1} + \frac{|W\beta_{i-1}|}{|W\beta_{i-1}| + |W\gamma_{i-1}|} \cdot b, \tag{4}$$

$$\gamma_i = W\gamma_{i-1} + \frac{|W\gamma_{i-1}|}{|W\beta_{i-1}| + |W\gamma_{i-1}|} \cdot b. \tag{5}$$

We succinctly represent the above equations as $\beta_i, \gamma_i = LinearDecomp(\beta_{i-1}, \gamma_{i-1})$, where $W$ and $b$ are understood to be tunable parameters of a neural network module.

Given the query, key, and value functions mentioned in Section 3.1 are linear transformations, the only module not accounted for in the context of transformers is the self-attention module. We handle the decomposition of the attention equations as follows:

$$\beta_{query}, \gamma_{query} = LinearDecomp(\beta_{in}, \gamma_{in}) \tag{6}$$

$$\beta_{key}, \gamma_{key} = LinearDecomp(\beta_{in}, \gamma_{in}) \tag{7}$$

$$\beta_{value}, \gamma_{value} = LinearDecomp(\beta_{in}, \gamma_{in}) \tag{8}$$

$$\beta_{attention} = \beta_{key}^T \beta_{query} / \sqrt{d_k} \tag{9}$$

$$\gamma_{attention} = ((\beta_{key} + \gamma_{key})^T (\beta_{query} + \gamma_{query}) / \sqrt{d_k}) - \beta_{attention} \tag{10}$$

$$\beta_{probs} = Softmax(Mask(\beta_{attention})) \tag{11}$$

$$\gamma_{probs} = Softmax(Mask(\beta_{attention} + \gamma_{attention})) - \beta_{probs} \tag{12}$$

$$\beta_z = \beta_{probs} * \beta_{value} \tag{13}$$

$$\gamma_z = (\beta_{probs} + \gamma_{probs}) * (\beta_{value} + \gamma_{value}) - \beta_z \tag{14}$$

$$\beta_{output}, \gamma_{output} = LinearDecomp(\beta_z, \gamma_z) \tag{15}$$

where above $Mask$ represents a causal mask in appropriate architectures and $Softmax$ is the softmax along the same dimension as in a standard implementation of self-attention.

Although in the original setting of CD, "input" and "output" refer to an input sequence $x$ and a model's output logits, the method generalizes to arbitrary *source* and *target* activations in the model. Furthermore, in the original context, the "relevant" portion of the decomposition was equal to 1 on a subsequence of the input and 0 elsewhere, but in a mechanistic interpretability context, the CD framework allows us to decompose activations into any choice of $\beta, \gamma$. Thinking of the network again as a computational graph, this gives us a method of calculating the effect of an arbitrary constituent of the activations of an arbitrary node on the activations of another node.

### 3.2.1 Choosing a decomposition of source nodes for mechanistic interpretability

In a manner analogous to the choice of ablation method elsewhere in mechanistic interpretability, the correct choice of starting decomposition can be subtle, and sometimes specific to the task. A general principle is that the relevant constituent of the activations at some node is the constituent that we hypothesize has a counterfactual effect on the task; often, this means the deviation from the mean activations over some distribution. (It follows that the irrelevant constituent of the activation at this node is the mean activation, which matches intuitions about mean-ablation.) We provide details on the evaluation tasks and name the distributions used for ablation in Appendix A.

## 3.3 Automating circuit discovery

### 3.3.1 Definitions of circuit discovery

Before delving into our method, we first formally define a *circuit*. To maintain consistency in notations with prior work (Shi et al., 2024; Elhage et al., 2021), we view a model as a computational graph $M$, where nodes are activations of the model components in its forward pass (e.g. attention heads and MLPs) and edges are the interactions between those components, and a circuit $C$ is a subgraph of $M$ responsible for certain behavior of interest. Naturally, the first step of circuit discovery is to define a *task* of interest to investigate. This requires specifying a dataset and a task-specific metric to measure the performance of $M$ and $C$, typically, by maximization of the task metric. Given an input $x$, similarly as to how the entire model defines a function $M(x)$ from inputs to logits, we also associate each circuit $C$ with a function $C(x)$, defined by ablating away the effect of all components in $M \backslash C$ (i.e. the components not included in $C$). Circuit discovery may be performed on a single input example or, likelier, the result may be averaged over multiple input examples to reduce variance. Typically we find averaging across 10-100 samples, depending on the tasks being studied, can yield stable performance. Detailed information about which models are studied, input sample counts, and how the task metrics are defined is found in Appendix A. Finally, note that our method finds the circuit of interest but does not automatically generate an interpretation of it, however, known interpretability techniques (Wang et al., 2023) can be used to do so.

### 3.3.2 Method

Here we present our algorithm for discovering circuits with CD-T. Although it is possible to perform CD at arbitrary granularity and on arbitrary components of a transformer (e.g., MLPs, query/key/value vectors), in this paper we focus solely on finding circuits consisting of attention heads in a transformer to be comparable to prior work, and our unit of analysis is either *the output of an attention head* or *the output of an attention head at a specific sequence position*. For clarity, our method focuses on including or excluding specific nodes from a computational graph, rather than edges; CD-T at once implicitly models both direct and indirect effects of one node to another, and our algorithm does not make a distinction between the two.

In each iteration, we search over potential source nodes to find the ones with highest relevance to a set of target nodes (initialized to the task objective). Once we find the set of nodes which are most relevant to the task, we prune them by some simple heuristic (in our experiments, simply greedily removing the nodes with the least magnitude of impact was often sufficient, or sometimes pruning was not necessary), and designate these most important nodes the target nodes for the next iteration of the algorithm. We halt when the collection of nodes achieves adequate performance (e.g, comparable to the whole network), or has not improved from the previous iteration.

We need to specify how "highest relevance to a set of target nodes" is determined. As described above, we start with a decomposition $\beta_s, \gamma_s$ of the output of a source attention head $s$, and we are interested in its relevance $R(s, T)$ to a set of target nodes $T$. Starting from $s$, we propagate the decomposition forward through the network using the CD-T equations for each module in the computational graph between $s$ and $T$; again, if there is some series of functions $f_n, ... f_1$, so that $T = f_n(f_{n-1}(...f_1(s)))$, and $f_1(\beta_s + \gamma_s) = \beta_1 + \gamma_1, f_2(\beta_1 + \gamma_1) = \beta_2 + \gamma_2, ..., f_n(\beta_{n-1} + \gamma_{n-1}) = \beta_T + \gamma_T$, we can straightforwardly apply the modules in order to obtain $\beta_T, \gamma_T$. In the case that $T$ is the output of the network, $\beta_T$ will have matching dimensions, and it is usually appropriate to let $R(s, T)$ be the task-specific objective as evaluated on $\beta_T$. (One intuitive instance of this is the case

---

**Algorithm 1** Building a circuit using CD-T for a specified task

---

**Input:** datapoint $x$, precomputed mean activations $\mu$
Denote $a_x(s)$ as the activation of $s$ on input $x$
Denote $\mu(s)$ as the precomputed mean activations of $s$ on an appropriate distribution
Initialize $C$ to store the circuit
Initialize $T$ to be the output of the model
**repeat**
    **for** each attention head $s$ upstream of $T$:
        Run the model from $x$ to $s$, and compute $\beta_s = a_x(s) - \mu(s), \gamma_s = \mu(s)$
        Propagate decomposition to target nodes with CD-T, obtaining $\beta_T, \gamma_T$
        Use $\beta_T, \gamma_T$ to compute the relevance of s to $T$, $R(s, T)$ using equation 16
    **end for**
    Heuristically pick the source nodes $s$ with the highest values of $R(s, T)$; define this set as $S$
    $C \leftarrow C \cup S$
    {# Refinement: Greedily prune away the unnecessary nodes}
    **repeat**
        **for** each node $n$ in $C$:
            $C^{'} \leftarrow C \backslash n$
            If $C^{'}(x) > C(x)$: $C \leftarrow C^{'}$
        **end for**
    **until** $C$ has not changed since the last iteration
    If $|C(x) - M(x)| < \epsilon$, return $C$ {# Circuit performs close to the full model}
    If $C(x)$ is not greater than last iteration, return $C$
    $T \leftarrow S$
**until** no upstream attention heads to $T$ are available
**return** $C$

---

where $T$ is a single "score" whose value we care about, so that $\beta_T$ is the contribution of $s$ to that score.) However, when $T$ is an arbitrary set of nodes in the network's internals, we must define a proxy metric for the relevance of $s$ to $T$; in this case, we define the relevance of a source node to a set of target nodes to be the sum of the relevances to the target nodes $t \in T$, and in this paper for $R(s, t)$ we choose a quick-to-compute measure of the size of $\beta_t$:

$$R(s, T) \coloneqq \sum_{t \in T} R(s, t) = \sum_{t \in T} \frac{\|\beta_t\|_{l_1}}{\|\gamma_t\|_{l_1}}. \tag{16}$$

Careful readers may have noticed that this particular metric measures the magnitude of one node's contribution to another, but not the "direction", so that nodes which maximize this metric may actually contribute negatively to the task under investigation. We found in our experiments that this does not preclude us from finding functioning circuits. On the contrary, this allows us to straightforwardly find nodes which have significant negative contribution to a task metric, such as the "Negative Name Mover Heads" in the IOI task. In principle, it is possible that different choices of target relevance metric can be developed to adjust the behavior of the circuit discovery algorithm to avoid (or accentuate) this behavior; we do not explore this in the paper but identify it as a possible avenue for further investigation. Our complete algorithm is described in Algorithm 1, presented in the specific case where we have chosen to decompose our source nodes $s$ so that $\beta_s$ is the activation's deviation from the mean over some distribution. More details on the heuristics used and a complexity analysis of the algorithm is found in Appendix B.

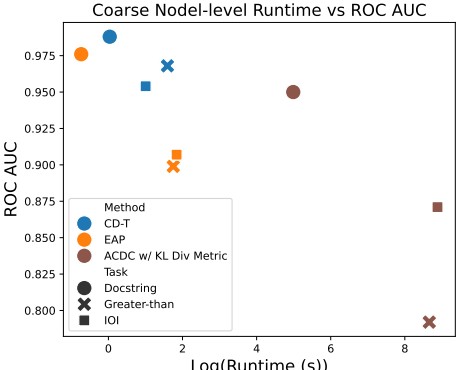 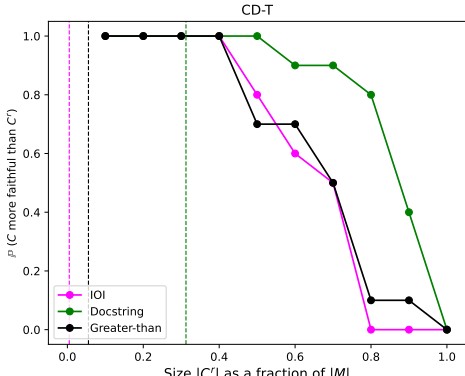

Figure 1: Left: Log algorithm runtime and ROC AUC comparison. Each dot represents an average measurement on one task with specific methods differentiated by colors.; Right: The relative faithfulness of CD-T circuits compared to random circuits from the reference distribution of varying sizes (x-axis). Dotted vertical lines indicate the actual size of the circuits. $C$ denotes a CD-T circuit. $C^r$ denotes a random circuit. $M$ denotes the full model.

## 4 EVALUATING CIRCUIT DISCOVERY ALGORITHMS

### 4.1 EXPERIMENTAL SETUP

We compare the performance of circuits recovered by CD-T with those obtained by two SOTA baselines: ACDC (Conmy et al., 2023) [3] and EAP (Syed et al., 2023) [4]. SAE-based methods (O'Neill & Bui, 2024; Marks et al., 2024; He et al., 2024) and classic neural network pruning methods are not considered here because they either require model training or focus more on compressing neural networks for faster inference to reduce storage requirements. Furthermore, pruning for interpretability is also shown to be outperformed by ACDC (Conmy et al., 2023). We evaluate the circuit performance on three tasks: indirect object identification (IOI) (Wang et al., 2023), greater-than (Greater-than) (Hanna et al., 2023), and docstring completion (Docstring) (Heimersheim & Janiak, 2023) (see Appendix A for details). They are three standard evaluation tasks to benchmark circuit discovery methods as prior work (Hanna et al., 2023; Wang et al., 2023; Heimersheim & Janiak, 2023) has *manually* found circuits in language models explaining for those tasks, which often serve as an imperfect reference standard for circuit discovery work (Conmy et al., 2023; Syed et al., 2023; O'Neill & Bui, 2024) due to the lack of ground-truth circuits. For both baselines, we perform node-level patching to be comparable since CD-T is a node-level patching technique. All experiments are conducted on an NVIDIA A100 GPU.

### 4.2 CD-T EXCELS IN IDENTIFYING THE CIRCUIT RESPONSIBLE FOR THE UNDERLYING ALGORITHM EFFICIENTLY

#### 4.2.1 RUNTIME AND THE RECOVERY OF MANUAL CIRCUITS

Leveraging the *manually discovered* circuits as a reference standard, we measure how much overlap there is between the recovered circuits and the reference circuits using ROC AUC. Specifically, we measure the ROC AUC by sweeping through a range of thresholds that determine cutoff scores to keep nodes in a circuit. For CD-T, we test by varying the percentile of top nodes to extract in every iteration, in the range of $[90, 99]$. At the same time, we also compute the average algorithm runtime across runs when sweeping through the thresholds for each method. Although CD-T supports arbitrary granular representations of the circuit, the runtime is measured by building a "coarse node-level" circuits by not splitting into sequence positions to be comparable with EAP and ACDC

---

[3]We adopt the ACDC optimized for KL diveregence instead of task-specific metrics because it's shown to perform better in Conmy et al. (2023).

[4]We consider EAP but not EAP-IG because EAP-IG performs comparably to EAP on the tasks we evaluate (Hanna et al., 2024), but with a runtime increase by a constant factor in theory.

because EAP doesn't support measuring node importance at sequence positions, and ACDC takes an impractical amount of time to achieve sequence-position splitting. To demonstrate CD-T's capability to identify circuits responsible for underlying algorithm for tasks with great efficiency, we plot the algorithm runtime against the ROC AUC and report the result in Figure 1-Left. From Figure 1-Left we can see CD-T outperforms the baselines by achieving high ROC AUC and low runtime at the same time. EAP obtains similar runtime as CD-T but performs slightly worse in recovering *manual* circuits for Greater-than and IOI. Even under this coarse nodel-level abstraction, ACDC is the least efficient as it often takes *hours* to identify a circuit, yet with no better ROC AUC on all tasks compared to other methods. To explain the better ROC AUC performance of CD-T, we find that CD-T is especially good at identifying negative heads and supporting heads, such as the negative name-mover heads and backup name-mover heads in IOI, which other algorithms struggle to do (more discussion in Appendix A).

### 4.2.2 PROBABILITY OF IDENTIFIED CIRCUITS TO BE MORE FAITHFUL THAN RANDOM CIRCUITS

Given a circuit and a task, one can evaluate how well the circuit performs on the task by measuring the task-specific metric, such as the logit difference on IOI. However, in mechanistic interpretability, instead of aiming to identify the best-performing circuits on the task-specific metric, an ideal circuit should be able to *faithfully replicate the full model behavior*. Following prior work (O'Neill & Bui, 2024; Marks et al., 2024), faithfulness is defined as the proportion of the full model's performance that a circuit explains, relative to the baseline performance when no specific input information is provided. Given a task, faithfulness is computed as $\frac{m(C)-m(\emptyset)}{m(M)-m(\emptyset)}$, where $m(C)$, $m(\emptyset)$, and $m(M)$ are the average of the task-specific metric performance over the dataset for the circuit, the corrupted model with all heads ablated, and the full model, respectively. To complement the imperfect nature of the *manual* circuits, leveraging faithfulness, here we provide another evidence to validate CD-T's mechanism perservation of the original model. We adopt a circuit hypothesis testing framework proposed in Shi et al. (2024) to test whether a circuit preserves the original model's performance by measuring the probability of a circuit $C$ to be more faithful to the original model $M$, compared to random circuits $C^r$ of the same size sampling from a reference distribution. This test verifies that the circuit is not a simple lucky draw from the distribution of random circuits, and ensures that it is better than at least a fraction $q^*$ of random circuits. In our experiment, we sample random circuits from all possible nodes in the full model, and we repeat the process for 10 times. Figure 1-Right shows the results of the best CD-T circuits (as measured by faithfulness and among different granularity) on the 3 tasks. Our best circuit for IOI is from attention heads at specific sequence positions, while for Greater-than and Docstring, they are from attention heads at a coarse level without the splitting. The results show CD-T circuits are 80% ($q^* = 0.8$) more faithful than random circuits of up to around 60% of the original model's size for IOI and Greater-than, and even those of up to 80% of the original model's size for Docstring, with just a small percentage of nodes in the CD-T circuits ($0.4\%$ of full model size for IOI, $5.5\%$ for Greater-than, and $31.3\%$ for Docstring). This finding suggests the faithfulness of CD-T circuits is not due to random chance, and that a more fine-grained abstraction of circuits sometimes is necessary to obtain better faithfulness with an extremely small and specific set of nodes, as in the IOI case.

### 4.3 IDENTIFIED CIRCUITS MATCH FULL MODEL PERFORMANCE

### 4.3.1 FAITHFULNESS OF CIRCUITS OF VARYING SIZES

To understand how faithfulness changes as the size of the circuits grows, we compute node importance before pruning for each method [5], and track faithfulness performance as we add circuit nodes in order of importance, while ablating all other nodes with corrupted activations (see Appendix A). We sample attention heads in the original model one by one in random order to construct the random baseline, and report the results in Figure 2. Since the goal of mechanistic interpretability is to replicate the original model's behavior, namely achieving faithfulness $= 1$ [6], with the smallest amount

---

[5] ACDC is omitted in this experiment because of the prohibitive amount of runtime in hours to obtain individual node importance.

[6] Faithfulness can go beyond 1 because there are both positively and negatively contributing heads in the model. The original model performance relies on a certain dynamic of interactions between them. When the

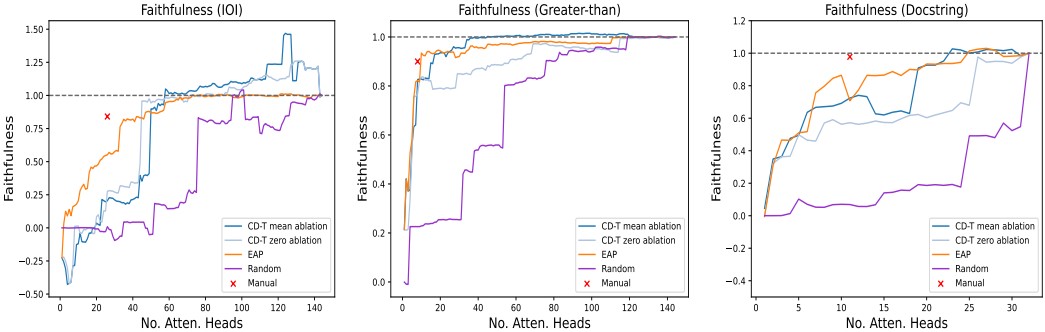

Figure 2: Faithfulness of CD-T circuits, EAP circuits, and randomly selected circuits of equivalent size for IOI, Greater-than and Docstring tasks. CD-T circuits obtains the full model's performance (faithfulness of 1) faster than EAP as attention heads are added in order of importance.

of nodes possible, this is naturally the first question we ask. First, we find CD-T circuits are able to perfectly replicate model behavior (faithfulness = 1) using fewer nodes than EAP for all the three tasks. Second, manual circuits outperform both CD-T and EAP of the same size, indicating room of improvement of advanced circuit discovery methods. Finally, we compare the effect of ablation methods on CD-T circuits' performance by including CD-T circuits obtained from zero-ablation in the plot, and find that mean-ablation tend to yield higher faithfulness compared to zero-ablation on Greater-than and Docstring, except for IOI where they are quite comparable. Mean-ablation is often viewed as a preferred way of ablation because it preserves a general concept of the task, hence is less destructive than zero-ablation. Similar finding is also discussed in other circuit discovery work (Conmy et al., 2023; Wang et al., 2023).

## 5 DISCUSSION AND LIMITATIONS

Despite having shown both qualitative and quantitative evidence of the strengths of CD-T in automated circuit discovery, our results is limited in scale and to the datasets and prior interpretation methods evaluated. More work is needed to generalize these findings to a broader set of models, datasets, and interpretation methods. Although our methods can be applied to construct circuits comprised of any of a neural network's internal components, we only discussed circuits built with purely attention heads. Automating the discovery of circuits with different/heterogeneous components, particularly the MLPs of a transformer, is a promising direction for further investigation. Finally, our method is itself limited in two ways: first, as mentioned in 3.3, the metric which measures the relevance of a source node to a target node does not differentiate signs of the contribution, which might be useful in an interpretability context. Second, while the algorithm is capable of finding circuits under reasonable default settings, some amount of effort is still applied to tune the heuristic elements, such as the number of heads to select in each iteration.

## 6 CONCLUSIONS

We have proposed contextual decomposition for transformers (CD-T), a set of mathematical decomposition principles to efficiently compute the contribution of model features to other model features within a transformer. CD-T can produce circuits of arbitrary granularity, and is the first able to produce circuits as fine-grained as attention heads at specific sequence positions while at the same time reduce runtime from hours to *seconds*, when it is compared with prior work (Syed et al., 2023; Conmy et al., 2023). CD-T improves upon limitations of prior work, as it is automatic and does not require any training or manually-crafted examples by humans. CD-T is fundamentally compatible with all common transformer architectures. On three standard circuit evaluation datasets (indirect object identification, greater-than comparisons, and docstring completion), we demonstrate that CD-

---

dynamic is not perfectly replicated and there's extra positively contributing effect, the circuit might give a higher prediction score than the original model would give, which results in a faithfulness $> 1$.

T outperforms ACDC and EAP by better recovering the manual circuits with an average of 97% ROC AUC under low runtimes. In addition, we provide evidence that faithfulness of CD-T circuits is not due to random chance by showing our circuits are 80% more faithful than random circuits of up to 60% of the original model size. Finally, we show CD-T circuits are able to perfectly replicate original models' behavior (faithfulness = 1) using fewer nodes than the baselines for all tasks. We hope that our future open-source implementation of CD-T benefits mechanistic interpretability research community by opening up a window from enabling more fine-grained analysis within a practical runtime.

## ACKNOWLEDGMENTS

GZ would like to thank Shawn Hu for valuable discussions that contributed greatly to mathematical formalization, experiment design, and important heuristic components of our method. We gratefully acknowledge partial support from NSF grant DMS-2413265, NSF grant 2023505 on Collaborative Research: Foundations of Data Science Institute (FODSI), the NSF and the Simons Foundation for the Collaboration on the Theoretical Foundations of Deep Learning through awards DMS-2031883 and 814639, NSF grant MC2378 to the Institute for Artificial CyberThreat Intelligence and OperatioN (ACTION), and a Berkeley Deep Drive (BDD) grant from BAIR and a Dean's fund from CoE, both at UC Berkeley.

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

# A   EXPERIMENT DETAILS

## A.1   TASK DESCRIPTION

Here we describe each task and how the experiments are performed in more detail; we also specify the exact model components which comprise the circuits we analyzed in the main paper. In this section, we use the convention of specifying attention heads in a model by the tuple (layer index, head index). For the IOI task, we add a third entry for output of the head at a specific sequence position, and follow their nomenclature for semantically labeling the sequence positions.

### A.1.1   INDIRECT OBJECT IDENTIFICATION (IOI) (WANG ET AL., 2023)

The indirect object identification (IOI) task is to predict the indirect object in a sentence with two entities, such as identifying "Mary" in the sequence "When Mary and John went for a walk, John gave an apple to ...". The objective originally defined for this task, which we also use, is the difference between the predicted logit of the indirect object (IO) token and the subject (S) token. Wang et al. (2023) carefully design their experiment and dataset code so that mean ablation occurs using the mean activations over the corrupted "ABC dataset", which replaces the three name tokens in the task with random names. Likewise, when setting the decomposition of a node, we set the "relevant" component to the deviation from the mean activations on the ABC dataset. This task was performed on GPT2-small. GPT-2 correctly performs this task about 99 percent of the time, so we did not take special measures to account for those samples where it doesn't in our circuit analysis. We identify circuits using 25 IOI samples drawn from mixed templates, and mean ablation is conducted using the corrupted ABC dataset. Another set of 100 IOI samples are used in evaluation. Manual circuit of IOI compared against in the ROC AUC experiment is: [(2, 2), (4, 11), (0, 1), (3, 0), (0, 10), (5, 5), (6, 9), (5, 8), (5, 9), (7, 3), (7, 9), (8, 6), (8, 10), (10, 7), (11, 0), (9, 9), (9, 6), (10, 0), (9, 0), (9, 7), (10, 1), (10, 2), (10, 6), (10, 10), (11, 2), (11, 9)], which is from Figure 2 in Wang et al. (2023).

**Details of Circuit Analysis**   For this task, in order to provide some intuitions about what CD-T calculates, we provide a number of heatmaps of the relevance scores at specific positions during various iterations of the circuit analysis. *In this section, we don't follow our automated circuit discovery algorithm exactly, but instead partially follow Wang et al. (2023)'s analysis to decide what sequence positions to search over and visualize.*

The first iteration of the algorithm finds the Name Mover Heads: (9, 9, end), (10, 0, end), and (9, 6, end); the Negative Name Mover Heads: (10, 7, end), (11, 10, end); and some Backup Name Mover Heads: (10, 2, end), (10, 6, end), (10, 10, end), described by Wang et al. (2023)

Following Wang et al. (2023)'s analysis further, and deviating from the normal course of our circuit-finding algorithm, we compute the relevance of nodes to the Name Mover Heads on just the end

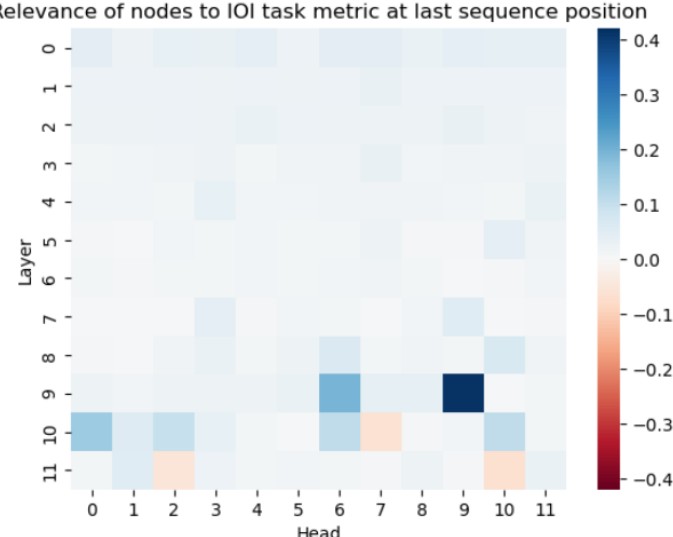

position, and find what they named the "S-Inhibition Heads", at (8, 10, end), (7, 9, end), and (7, 3, end), though there are some other relevant-looking contenders:

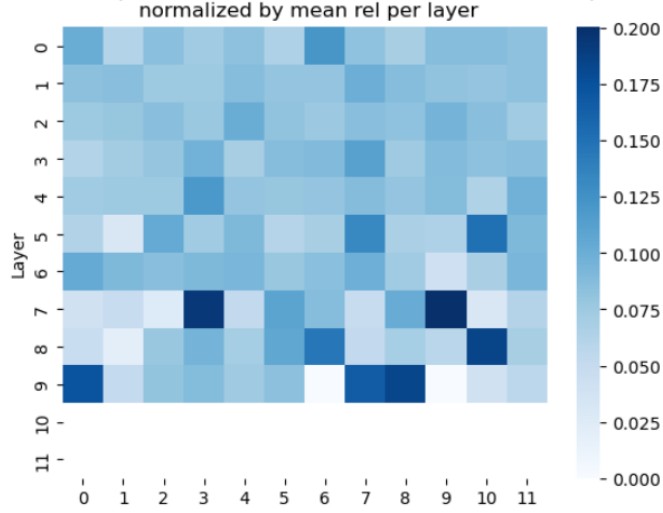

Continuing to follow their analysis, Wang et al. (2023)'s analysis further, we compute the relevance of nodes to the S-Inhibition Heads at the S2 position, and find our first minor disagreement with their process: though we find two of what they named the Induction Heads at (5, 5, S2), (5, 8, S2), (5, 9, S2) we don't find the one at (6, 9, S2) but instead find one at (5, 10, S2), and find a head their later analysis called a Duplicate Token Head, at (3, 0, S2).

Next we compute the relevance of nodes to the Induction Heads at the S2 position, and find that (3, 0, S2) mostly drowns out the signal of the other Duplicate Token Heads:

Finally, we compute the relevance of nodes to the Induction Heads at the S1+1 position, and find (4, 11, S1+1), with the other Previous Token Head they found at (2, 2, S1+1) a top contender, though there are other heads not accounted for in their analysis:

Overall, there is significant but not perfect agreement with the results and analysis of the IOI paper. We also attempted analysis by computing relevance to intermediate matrices (i.e., the key, query,

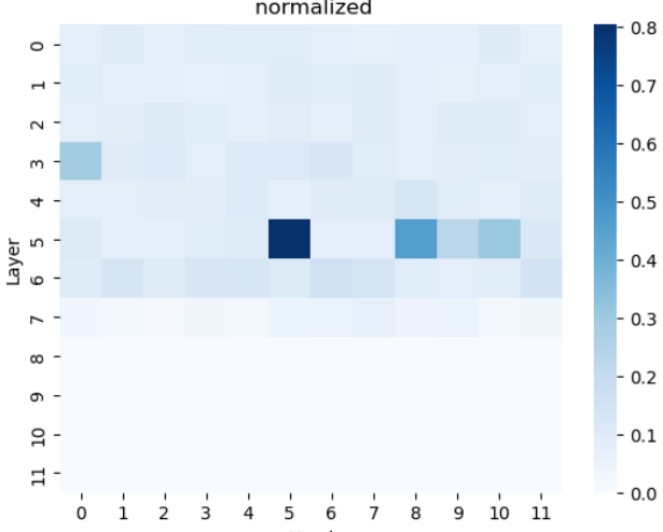

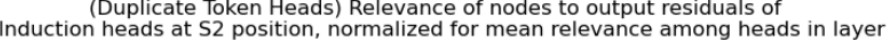

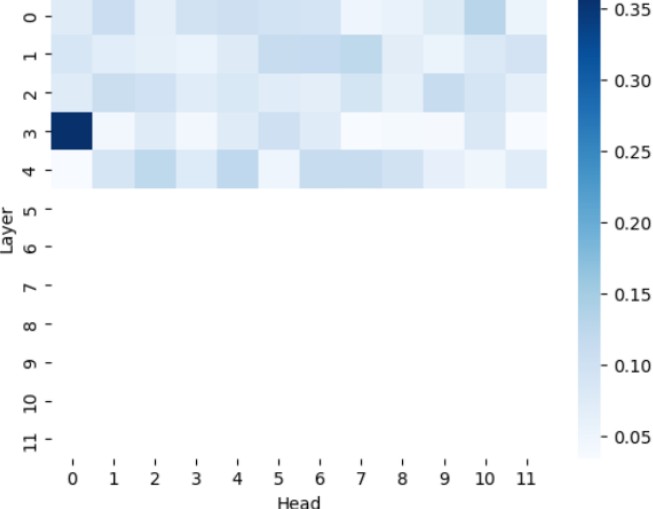

value vectors) in the attention calculation, but found the plots to be qualitatively similar, possibly due to the fact that CD-T by design propagates relevances from these vectors to the attention head outputs as well.

Another fact of note is that the scales on these plots vary substantially. It would be desirable to find an interpretable normalization method to put these on the same scale with some intrinsic meaning, or otherwise explain the causes of this phenomenon.

### A.1.2 GREATER-THAN (HANNA ET AL., 2023)

The Greater than task is to predict the last two digits in an incomplete sentence following the template "The [noun] lasted from the year XXYY to the year XX". And we expect the model to assign higher probability to years greater than YY. The objective originally defined for this task, which we also use, is the sum of probabilities assigned to tokens corresponding to greater years, minus the sum of probabilities assigned to tokens corresponding to lesser years. (Some probability is assigned to tokens which don't correspond to numbers at all.) For our "mean-ablation", we simply take the

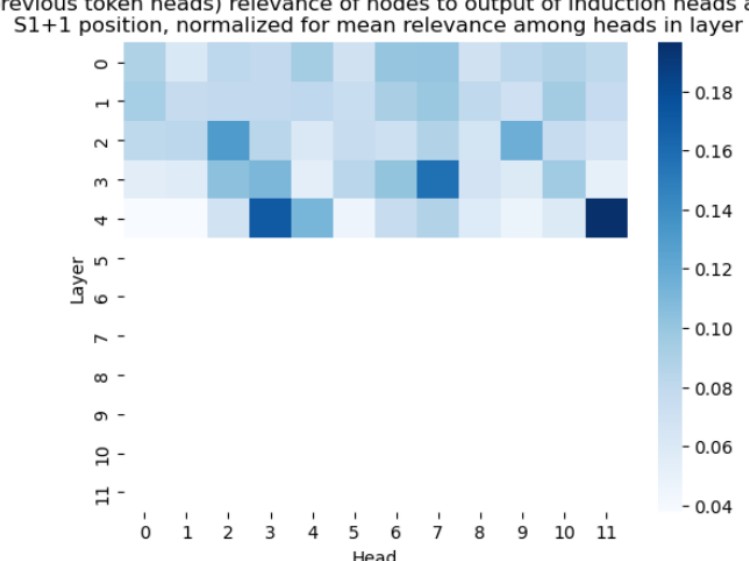

mean over the activations over 100 negative datapoints (impossible completions, with the ending year preceding the starting century), and as above, when setting the decomposition at a source node, define the relevant component to be the deviation from the mean activation over this distribution. This task was performed on GPT2-small. GPT-2 correctly performs this task about 99 percent of the time, so we did not take special measures to account for those samples where it doesn't in our circuit analysis. We identify circuits using a random sample of 100 datapoints provided by Hanna et al. (2023), and mean ablation is conducted using the negative impossible completion samples. Another set of 100 samples are used in evaluation.

It should be noted that our result and the results in Hanna et al. (2023) are not directly comparable, since they also attempt to investigate the influence of MLPs and their resulting circuit removes most of the MLPs in GPT-2. For the comparisons found in the main paper, we have compared our result (with all MLPs) to the circuit which is obtained by taking all the attention heads named in Hanna et al. (2023) (also with all MLPs), which is a circuit distinct from the one found in their work.

Manual circuit of Greater-than compared against in the ROC AUC experiment is: [(5, 1), (5, 5), (6, 1), (6, 9), (7, 10), (8, 8), (8, 11), (9, 1)].

Independently of the comparison, it remains true that keeping the relatively small proportion of attention heads in our circuit results in recovering almost all of GPT-2's capability on this task; see below.

|  | Task-specific metric | Correct guess rate |
| --- | --- | --- |
| Full model | 0.817 | 0.992 |
| All attention heads ablated | -2.095 | 0 |
| Their circuit | 0.768 | 0.989 |
| Their circuit (most MLPs ablated) | 0.727 | N/A |
| Our circuit | 0.761 | 0.981 |

### A.1.3 DOCSTRING (HEIMERSHEIM & JANIAK, 2023)

The goal of the Docstring task is to predict the next variable name in a Python docstring. For example, given a function with variable names LOAD, SIZE, FILES, and LAST, the task is to predict the word after one :PARAM. According to docstring conventions, this should be the variable name in the function definition which follows the most recent variable name to appear after :PARAM. The objective originally defined for this task is the logit assigned to the correct variable name, minus the max logit assigned to all other variable name tokens found in the function signature. For

our "mean-ablation", we use their `random_random` dataset which randomize both the variable names in the function definition and in the docstring of prompts, and correspondingly, when setting the decomposition at a source node, define the relevant component to be the deviation from the mean activation over this distribution. We identify circuits using a 100 datapoints sampling for the dataset provided by Heimersheim & Janiak (2023), and mean ablation is conducted using the corrupted `random_random` dataset. Another set of 100 samples are used in evaluation. This task was performed on a 4-layer attention-only transformer trained on natural language and Python code (`attn-only-4l`) released with the TransformerLens library for the express purpose of facilitating mechanistic interpretability research. Another complication is that the toy model only guesses the correct token between 60 and 65 percent of the time. To account for this, we perform our circuit analysis and evaluation on the subset of input examples for which the model performs the task correctly.

Our circuit consists of the nodes (3, 0), (3, 6), (1, 4), (0, 5), (0, 0), (1, 0), (1, 4), (0, 1), (2, 3), (1, 2). Heimersheim & Janiak (2023) find the circuit (0, 2), (0, 4), (0, 5), (1, 2), (1, 4), (2, 0), (3, 0), (3, 6) with their initial analysis, and heuristically observe that three heads help to obtain the "augmented circuit" (0, 2), (0, 4), (0, 5), (1, 2), (1, 4), (2, 0), (3, 0), (3, 6), (1, 0), (0, 1), (2, 3).

Manual circuit of Docstring compared against in the ROC AUC experiment is: [(0, 2), (0, 4), (0, 5), (1, 2), (1, 4), (2, 0), (3, 0), (3, 6)].

|                          | Task-specific metric | Correct guess rate |
|--------------------------|:--------------------:|:------------------:|
| Full model               | 3.661                | 1.0                |
| All attention heads ablated | -2.095            | 0                  |
| Their circuit            | 2.884                | 0.54               |
| Augmented circuit        | 3.524                | 0.66               |
| Our circuit              | 3.235                | 0.57               |

# B ALGORITHM DETAILS

## B.1 HEURISTICS

In addition to the greedy pruning presented in Algorithm 1 as a refinement step, here we describe other heuristic elements to the algorithm.

- The threshold for determining which set S of highest-contributing nodes may be varied: it is possible to pick the top $N$ nodes or fraction of nodes, or to automatically detect outliers. Empirically, we find that the distribution of node contributions varies quite severely, so finding a heuristic which works in all cases is actually an object of future work. In this paper, we set the threshold by varying the percentile of top nodes to extract, in the range of [90, 99] to obtain the ROC AUC in section 4.2.1.

- Empirically we find, for a fixed target node, the relevance scores of source nodes in different layers to this target node may have different expected magnitudes, due to the numerical effects of propagation through the network. To account for this, we normalize the relevance scores by dividing by the average magnitude across scores found in a given layer.

- To avoid the risk of propagation equations becoming numerically unstable after a large number of iterations, especially if the relevant and irrelevant constituents differ in sign at a specific index, we set the value of one of rel/irrel at this position to 0 and the other to the sum of the two terms.

## B.2 COMPLEXITY ANALYSIS

To provide more clarity, the computational complexity of the algorithm satisfy the following properties:

- Each decomposition (of a set of target nodes with respect to a set of source nodes) requires cost in FLOPs similar to one forward pass of the model (and often less, since values prior

to the source nodes can be cached and values after the target nodes do not need to be calculated).

- The core of the algorithm is a loop, where in each iteration, we search over all nodes which can potentially have high relevance to the target nodes. (This means that heuristically excluding some nodes from the search can potentially significantly decrease cost in FLOPs, and cost in FLOPs increases linearly with respect to the granularity with which we separate the nodes in the model.)

- The memory footprint of a single decomposition, including the forward pass, is a small (less than 3) constant multiple of the cost of a forward pass; the only added costs are to keep track of the relevant and irrelevant constituent tensors separately, as well as bookkeeping of components of the target decomposition metric.

- The cost (in FLOPs or memory) of performing the analysis on a set of input examples is linear in the number of input examples, since the same set of computations needs to be done with respect to each example.

