# OpenReview forum: "Efficient Automated Circuit Discovery in Transformers using Contextual Decomposition"
_ICLR.cc/2025/Conference — ICLR 2025 Poster_

### Official Review · Reviewer_Lhid · 2024-11-01

**Soundness:** 3
**Presentation:** 4
**Contribution:** 3
**Rating:** 8
**Confidence:** 3

**Summary:**

The paper proposes an automatic, mechanistic interpretability approach for finding circuits in transformer models. Their approach propagates backwards from the model output, finding a circuit of nodes most relevant to the given output, and from there the nodes most relevant to those nodes, and so on. These circuits are task-specific subgraphs in the larger model compute graph that are most stimulated (and most relevant) to the input type or task under consideration.

The authors consider circuit discovery for the attention heads of transformer models, choosing to leave the MLP layers in an attention block for later work. Their linear decomposition model is mechanistic in that it solely depends on the weights and biases of each layer of the network and the relevance metric calculated for the previous layer. It does appear to be data-dependent in that it still relies on a dataset for calculating these relevance metrics. It also relies on precomputed mean activations to do so.

**Strengths:**

1) The paper's originality is in applying a novel formulation of an existing method - Context Decomposition - to transformer models, albeit to the attention heads of the transformer models. While novel, I am not in the space of circuit discovery in neural networks and unable to judge how novel this is.
2) The presentation is clear and the limitations in data dependence and precomputation are made clear to the reader in algorithm pseudocode, which is also concise and clear. Experimentation is presented to clearly establish the work's superior performance over the baselines. Notation remains consistent through the paper to my eyes.
3) Significance: The work presents the an automatic circuit discovery approach using CD (extending it as CD-T) in transformer models, and clearly outperforms its chosen baselines. The approach is moreover mechanistic and iterative rather than probabilistic and therefore also comes with a guarantee of circuit discovery given certain assumptions on available data and time.

**Weaknesses:**

1) I am unsure of the significance of the work due to the iterative nature of the process and the need for data to adequately stimulate different 'tasks' (or behavior modes, perhaps?) and therefore discover circuits in the model. This is to my eyes not clearly explained. [Addressed]
2) Prior work is noted to have conducted circuit discovery on large language models, and I can't easily find the models used for circuit discovery here. That lack of clarity is concerning, especially when using the manually discovered circuits as a target and prior art as baselines. [Addressed]
3) I am also uncertain of scalability of this approach, considering that the network architectures under evaluation are not clear and that it has been restricted to attention heads rather than MLP layers. I am also uncertain of the effects of sequence length, autoregressive or non-autoregressive decoders. Section 5 does not make these things clear. [Addressed in part; prior art remains deficient as well here - future work?]

**Questions:**

1) I would suggest clearly presenting model architectures under evaluation and assumptions being made on input data for the approach - the mechanistic relevance metrics are ultimately calculated from the outputs derived from input data.
2) How would this scale to large training datasets in LLMs? Is there a minimum dataset required? Can the authors comment on this?
3) What is the effect of autoregressive models and sequence to sequence decoding on this approach? Is there a difference? I don't see a clear analysis in Section 5, and while that is not necessary it would be nice to have the authors comment on this in Section 6.
Questions addressed in comments.

---

> ### Author Response · Authors · 2024-11-22
>
> Thanks for the helpful comments and thoughtful feedback (and large time commitment). We sincerely appreciate your critique and address your concerns below.
>
> > I am unsure of the significance of the work due to the iterative nature of the process and the need for data to adequately stimulate different 'tasks' (or behavior modes, perhaps?) and therefore discover circuits in the model. This is to my eyes not clearly explained.
>
> Thank you for pointing this out! Regarding the need of data, this is really an indispensable part in circuit discovery in general for mechanistic interpretability. The definition of circuit discovery is to find a circuit that a model uses to perform a specific task, and a task has to be defined by a specified dataset and a corresponding task metric. In other words, circuit discovery itself cannot be meaningful without data. Regarding the concern about the iterative nature of CD-T, to our knowledge, if one wants to form a circuit through exact calculation (i.e. to avoid inducing estimation errors or manually crafting samples to train a model), all existing state-of-the-art methods available are iteration-based [4,5]. Among all existing iteration-based methods, CD-T is shown in the paper that is able to scale the best in an LLM (GPT2-small) by achieving the lowest runtime and the best performance.
>
> > Prior work is noted to have conducted circuit discovery on large language models, and I can't easily find the models used for circuit discovery here. That lack of clarity is concerning, especially when using the manually discovered circuits as a target and prior art as baselines.
>
> Sorry for the confusion! This information was actually included in Appendix A in our submission. We test our algorithm on GPT-2-small for the IOI and Greater-than tasks, and on a 4-layer attention-only transformer for the Docstring task. These choices of models are the same as what were used in their original papers [1-3] to be comparable. Section 3.3.1 (line 285-286) has also been adjusted to clarify.
>
> > I am also uncertain of scalability of this approach, considering that the network architectures under evaluation are not clear and that it has been restricted to attention heads rather than MLP layers. I am also uncertain of the effects of sequence length, autoregressive or non-autoregressive decoders. Section 5 does not make these things clear.
>
> Please see our response above regarding the model architectures. CD-T itself is a general method that can be used to calculate the effect of arbitrary constituent of the activations of an arbitrary node on the activations of another node (line 258-260). Hence the user can easily decide any compositions of nodes (e.g, any combinations of attention heads, MLPs, or even the layer norm modules) to look at using this tool. In the experiments, we chose to focus on examining circuits purely of attention heads to be comparable with prior work [1-3], because a lot of them didn’t include MLPs and only focused on explaining attention heads in the found circuits. We have modified line 293-294 in section 3.3.2 to clarify. We didn’t observe any impact on the resulting circuits due to a difference in the sequence length. A discussion on the difference between the two types of models is included in a separate response below.
>
> > I would suggest clearly presenting model architectures under evaluation and assumptions being made on input data for the approach - the mechanistic relevance metrics are ultimately calculated from the outputs derived from input data.
>
> Thank you for the suggestion! Please see our response above regarding the model architectures. To our knowledge, all existing circuit discovery methods, including our CD-T, don’t make any assumptions on the input data. In the circuit discovery setting, we only require users to specify a dataset that they are interested in interpreting a model’s behavior for, and a corresponding task metric to measure the performance. Section 3.3.1 (line 277-279) has been adjusted to make this fact clearer.

---

> ### Author Response · Authors · 2024-11-22
>
> > How would this scale to large training datasets in LLMs? Is there a minimum dataset required? Can the authors comment on this?
>
> This is a great question! In all the three dataset we evaluated [1-3], the authors define a narrow task, and then construct a dataset where the correct behavior is clearly known to elicit the circuit needed to “perform the task”. As such, our method naturally can be performed on a single inference datapoint, but to answer the question of “what circuit performs the task in general”, it is better to perform it on the mean activations over 10-100 example points to reduce variance in the result. Details on how to handle sequence position information to align different examples are found in [1]. In our experiments, to be comparable, we used the same example counts as prior work [1-3] to obtain the circuits, with the exact numbers included in Appendix A. Typically we, alongside most prior work [1-3], find 10-100 datapoints is a sufficient range to obtain stable circuit discovery results in LLMs. More datapoints don’t necessarily yield better performance. Section 3.3.1 (line 282-285) has also been adjusted to make this clearer.
>
> > What is the effect of autoregressive models and sequence to sequence decoding on this approach? Is there a difference? I don't see a clear analysis in Section 5, and while that is not necessary it would be nice to have the authors comment on this in Section 6.
>
> Thank you for the suggestion! The difference in CD-T implementation for the two types of models is in enforcing different causal masks as mentioned in line 252. Although our CD-T code is designed and implemented to be run on both types of models, in the experiments shown in the paper, we only focused on generative models (GPT2-small and a 4-layer attention-only transformer) so we have prior work as a reference standard when quantifying our results. We couldn't find any prior work performing circuit discovery on encoder-based models, nor any work comparing the circuits distilled from the two kinds of models given the same task. With the flexibility of CD-T, this might be an interesting future direction to explore.
>
> Reference:
>
> [1] Interpretability in the Wild: a Circuit for Indirect Object Identification in GPT-2 small. Wang et al. 2023.
>
> [2] How does GPT-2 compute greater-than?: Interpreting mathematical abilities in a pre-trained language model. Hanna et al. 2023.
>
> [3] A circuit for Python docstrings in a 4-layer attention-only transformer. Haimersheim & Janiak. 2023.
>
> [4] Towards automated circuit discovery for mechanistic interpretability. Conmy et al. 2023.
>
> [5] Attribution patching outperforms automated circuit discovery. Syed et al. 2023.

---

> ### Comment · Reviewer_Lhid · 2024-12-02
> **Review Response**
>
> In response to review comments from the authors, I am editing my review above. The authors have adequately addressed my questions.

---

### Official Review · Reviewer_tnfo · 2024-11-03

**Soundness:** 3
**Presentation:** 3
**Contribution:** 3
**Rating:** 6
**Confidence:** 3

**Summary:**

The paper introduces Contextual Decomposition for Transformers (CD-T) for building interpretable circuits in llms for better mechanistic interpretability in transformers efficiently. In contrast to other methods, CD-T employs a mathematical decomposition that isolates the contributions of specific features, allowing CD-T to discover circuits at various levels of granularity. Results show significant improvements in runtime and interpretability over existing methods such as ACDC and EAP on several tasks.

**Strengths:**

1.	The paper is relatively well presented. It provides a clear description of the method, experiment details, and contribution.
2.	Using the decomposition approach for designing/scaling LLM models is relatively novel and effective. Results show promising improvements in computational efficiency while maintaining circuit quality, and the method seems to be agnostic to the specific types of transformer models.

**Weaknesses:**

1.	The description of the algorithm can be improved. In particular, the algorithm description is relatively informal and could benefit from more details and formalism. For “Prune nodes from S for which doing so increases the task metric”, how is increasing the task metric evaluated?

**Questions:**

1. See W1; In addition, what’s the computational complexity of the algorithm in terms of the input parameter?
2. How does pruning affect the mechanistic interpretability of the model?

---

> ### Author Response · Authors · 2024-11-22
>
> Thanks for the helpful comments and thoughtful feedback (and large time commitment). We sincerely appreciate your critique and address your concerns below.
>
> > The description of the algorithm can be improved. In particular, the algorithm description is relatively informal and could benefit from more details and formalism. For “Prune nodes from S for which doing so increases the task metric”, how is increasing the task metric evaluated? In addition, what’s the computational complexity of the algorithm in terms of the input parameter?
>
> Thank you for the suggestion! We’ve revised Algorithm1 to make it more precise, which also includes details on how exactly we pruned the nodes. Text in section 3.3.1 has been also adjusted to include more details to support the algorithm. In short, if removing a node in a circuit results in an improvement (usually means a “greater” score in the maximization setting, which is true in all the 3 tasks we evaluated) in the score measured using the task metric compared to the original circuit before the removal, we deem this head to be unnecessary and remove it. The computational complexity of the algorithm scales linearly in the number of examples. We’ve also added in a detailed discussion on heuristics used and a complete complexity analysis in Appendix B.
>
> > How does pruning affect the mechanistic interpretability of the model?
>
> In our paper, pruning refers to removing nodes from a candidate circuit, and not to the model as a whole. In general, our method only finds circuits and does not attempt to perform the interpretation of the circuits, which is still a task left to the user. Some of the papers we use as our baselines have performed much of this interpretation analysis [1-3] to characterize functionality of attention head groups in a circuit. After finding a circuit with high performance, the user can then use the existing techniques to find an interpretation of the resulting circuit as well as to interpret the contribution of the pruned node. There is no reason in principle that the resulting pruned circuit cannot be interpreted in the usual way.
>
> Reference:
>
> [1] Interpretability in the Wild: a Circuit for Indirect Object Identification in GPT-2 small. Wang et al. 2023.
>
> [2] How does GPT-2 compute greater-than?: Interpreting mathematical abilities in a pre-trained language model. Hanna et al. 2023.
>
> [3] A circuit for Python docstrings in a 4-layer attention-only transformer. Haimersheim & Janiak. 2023.

---

> > ### Comment · Reviewer_tnfo · 2024-11-25
> >
> > Thanks for the clarification. The rebuttal has addressed all of my concerns.

---

### Official Review · Reviewer_ZeEK · 2024-11-04

**Soundness:** 3
**Presentation:** 3
**Contribution:** 3
**Rating:** 5
**Confidence:** 3

**Summary:**

This paper introduces a novel method CD-T that leverages contextual decomposition for mechanistic interpretability in transformers. CD-T's granularity is fine-grained up to attention heads (does not include MLP layers). They report the results of their method (classification metrics and faithfulness of the circuit as compared to randomly sampled circuits) for three tasks, namely indirect object identification, greater-than comparisons, and docstring completion.

**Strengths:**

1. Motivation, relevant scientific terms and prior works are well-written, making the paper accessible to researchers who are non-experts in this area.

2. The proposed method can be easily used for all transformer architectures (encoder-decoder and decoder-only, as per line 72). Further, the proposed method significantly reduces the runtime required for inference.

**Weaknesses:**

1. Manual circuits are not fully explained (definition, how they are computed, cost of computation, etc.) - manual circuits are used as reference during evaluation (line 405), and it is necessary to provide these details regarding them.

2. It is unclear how CD-T works for inference - is there a distinct circuit discovered for each inference datapoint, or is the circuit found on a broader scale, so as to tradeoff size vs. performance for a larger test set?

3. The authors are encouraged to discuss the practical utility of CD-T. Once their method yields a circuit, could additional analyses, such as on attention heads, be conducted? For instance, are there heads that consistently produce positive or negative outputs across all data points, or do the importance of attention heads vary based on input data characteristics?

4. What transformer architecture is used for the experimental results?

**Questions:**

Line 20: "CD-T is the first ---?" there seems to be a missing word here

---

> ### Author Response · Authors · 2024-11-22
>
> Thanks for the helpful comments and thoughtful feedback (and large time commitment). We sincerely appreciate your critique and address your concerns below.
>
> > Manual circuits are not fully explained (definition, how they are computed, cost of computation, etc.) - manual circuits are used as reference during evaluation (line 405), and it is necessary to provide these details regarding them.
>
> Sorry for the confusion! Details of the evaluated manual circuits were actually included in Appendix A. During this revision, we have additionally put in sentences in the main text to explain the manual circuits (line 58-59, line 119-121). To answer your question in full, the “manual” circuits we evaluated precede the existence of any “automated” circuit discovery algorithm, and the term “manual circuits” reflects that a substantial amount of direct human effort went into isolating the circuits that the papers [1-3] produce.
> All of our “manual circuits” baselines are found by either manually examining the attention patterns of different attention heads or conducting ad-hoc task-specific analyses to decide the connectivity of a circuit. As such, there is no clear notion of runtime.
> Unlike the focus of our paper, optimizing for a more efficient algorithm to obtain a circuit, those papers tried to answer different questions, through manual efforts, 1. Whether it is possible to find a circuit to explain a model’s behavior performing a given task and 2. Interpreting each attention heads’ functionality in the circuit.
>
> > It is unclear how CD-T works for inference - is there a distinct circuit discovered for each inference datapoint, or is the circuit found on a broader scale, so as to tradeoff size vs. performance for a larger test set?
>
> Apologies! In all the three dataset we evaluated [1-3], the authors define a narrow task, and then construct a dataset where the correct behavior is clearly known to elicit the circuit needed to “perform the task”. As such, our method naturally can be performed on a single inference datapoint, but to answer the question of “what circuit performs the task in general”, it is better to perform it on the mean activations over 10-100 example points to reduce variance in the result. Details on how to handle sequence position information to align different examples are found in [1]. In our experiments, to be comparable, we used the same example counts as prior work [1-3] to obtain the circuits, and the exact numbers were included in Appendix A. Section 3.3.1 (line 282-285) has also been adjusted to make this clearer.
>
> > The authors are encouraged to discuss the practical utility of CD-T. Once their method yields a circuit, could additional analyses, such as on attention heads, be conducted? For instance, are there heads that consistently produce positive or negative outputs across all data points, or do the importance of attention heads vary based on input data characteristics?
>
> Thank you for the suggestion! Some of the papers we use as our baselines have performed much of this interpretation analysis [1-3] to characterize functionality of attention head groups in a circuit. Since we are attempting to evaluate the same model on the same task, the question of whether specific nodes help or hurt task performance will necessarily be the same. Instead, our paper’s focus is on efficiently finding the circuit which emulates the original model’s performance, at which point known techniques can be used to interpret the found circuits. Section 3.3.1 (line 286-288) has been updated to make the nature of our contribution clearer.
>
> > What transformer architecture is used for the experimental results?
>
> Sorry for the confusion! This information was actually included in Appendix A in our submission. We test our algorithm on GPT-2-small for the IOI and Greater-than tasks, and on a 4-layer attention-only transformer for the Docstring task. These choices of models are the same as what were used in their original papers [1-3] to be comparable. Section 3.3.1 (line 285-286) has also been adjusted to clarify.
>
> > Line 20: "CD-T is the first ---?" there seems to be a missing word here
>
> Thank you for pointing out! This sentence has now been fixed.
>
> Reference:
>
> [1] Interpretability in the Wild: a Circuit for Indirect Object Identification in GPT-2 small. Wang et al. 2023.
>
> [2] How does GPT-2 compute greater-than?: Interpreting mathematical abilities in a pre-trained language model. Hanna et al. 2023.
>
> [3] A circuit for Python docstrings in a 4-layer attention-only transformer. Haimersheim & Janiak. 2023.

---

### Meta-Review · Area_Chair_cGea · 2024-12-20

**Metareview:**

Thank you for your submission to ICLR. This paper presents CD-T (contextual decomposition for transformers), a method which aims to build interpretable circuits in large language models. It computes these circuits efficiently, with possible reductions in runtime from hours to seconds, in comparison with existing baselines. In addition, CD-T performs well empirically, compared to baselines, on multiple circuit evaluation datasets.

Reviewers appreciate the clarity of presentation, the method novelty, and the performance of the method. While there was some concern about certain relevant background descriptions and definitions, as well as the method’s scalability, the authors eventually addressed all of the reviewers’ concerns.

**Additional Comments On Reviewer Discussion:**

During the rebuttal period, the authors answered all of the reviewers’ questions thoroughly. Reviewers, on the whole, appreciated these updates and felt their concerns were addressed, which led to an increase in scores.

---

### Decision · Program_Chairs · 2025-01-22

Accept (Poster)